# Evaluation of the Efficacy of Lacer Hali^TM^ Treatment on the Management of Halitosis: A Randomized Double-Blind Clinical Trial

**DOI:** 10.3390/jcm10112256

**Published:** 2021-05-23

**Authors:** Laiqi Xiang, Rosa Rojo, Juan Carlos Prados-Frutos

**Affiliations:** 1Doctoral Program in Health Sciences, Faculty of Health Sciences, Rey Juan Carlos University, Avenida Atenas s/n, 28922 Alcorcón, Madrid, Spain; laiqi1112@gmail.com; 2Practice of Dentistry and Halitosis at Core Centro Dental, 28001 Madrid, Spain; 3Faculty of Dentistry, Alfonso X el Sabio University, Villanueva de la Cañada, 28691 Madrid, Spain; 4Department of Medicine Specialties and Public Health, Faculty of Health Sciences, Rey Juan Carlos University, Avenida de Atenas s/n, 28922 Alcorcón, Madrid, Spain; juancarlos.prados@urjc.es; 5IDIBO Group, Health, IDIBO Group (Research, Development and Innovation Group in Dental Biomaterials), Rey Juan Carlos University, Avenida de Atenas s/n, 28922 Alcorcón, Madrid, Spain

**Keywords:** halitosis, oral malodor, octenidine, Lacer Hali^TM^, organoleptic score, OralChroma^TM^

## Abstract

Background: Halitosis of oral origin is very common in the general population. Due to their antimicrobial properties, chlorhexidine-based products are widely used in the management of this condition, but these are associated with reversible side effects. In this study we evaluated the efficacy of Lacer Hali^TM^ mouthrinse and toothpaste in subjects with intraoral halitosis after several applications under normal conditions of use. Methods: In this randomized clinical trial with mouth rinse and toothpaste, single-center, double-blinded, parallel participants were assigned to an experimental group (Lacer Hali^TM,^, *n* = 20), a positive control group (Halita^TM^, *n* = 20), and a placebo group (*n* = 20). The active duration of the study was 18 days. The clinical follow-up evaluations were performed at five time points (T0, T1, T2, T3, and T4). The intensity of halitosis was evaluated by organoleptic measurement and the portable gas chromatograph OralChroma^TM^. The data were analyzed using generalized mixed linear models. Results: Sixty patients completed the study. Lacer Hali^TM^, in comparison with Halita^TM^, did not show statistically significant differences at any time during the study except for the levels of hydrogen sulfide and total volatile sulfur compounds at 15 days, where Halita^TM^ was better. Compared to the placebo treatment, Lacer Hali^TM^, was significantly more efficient, in terms of both the organoleptic evaluations at 8 days and the levels of hydrogen sulfide. Conclusions: Lacer Hali^TM^ is an alternative to chlorhexidine-based toothpaste and mouthwashes in the management of halitosis.

## 1. Introduction

Halitosis, or oral malodor, is defined as the set of unpleasant odors emitted from a person’s mouth. Halitosis of oral origin is very common in the general population, representing 31.8% [1]. It lowers quality of life, and is a cause of significant social and psychological disability [2].

Not all halitosis is of oral origin—extraoral halitosis is caused by problems in the respiratory tract, disorders of the gastrointestinal tract, some systemic diseases, or metabolic disorders, and, less frequently, by carcinomas [3,4]. However, approximately 90.0% [5] of all halitosis cases are caused by the accumulation of bacteria and food residues present at the tongue and periodontal locations [6], even in individuals free of periodontal diseases such as gingivitis or periodontitis [7].

As intraoral halitosis is mainly produced by bacterial colonization, it is associated with inflammation and some infectious conditions of soft oral tissues [8,9]. Rarely, halitosis could be a sign of several other diseases such as oral cancer, which require a rapid diagnosis [10].

Treatment can be addressed by mechanical interventions such as brushing, flossing, tongue scraping [11], and chemical antibacterial agents such as chlorhexidine (CHX), phenol, tricolosan, dioxide chlorine, alcohol, and metal ions [12,13].

The most common methods used to diagnose intraoral halitosis are organoleptic evaluation and gas chromatography to quantify volatile sulfur compounds (VSCs).

Organoleptic measurement is considered the gold standard in detecting halitosis despite its subjectivity [14,15]. However, VSC analysis is more reliable to detect halitosis in the case of OralChroma™ (Abimedical, Kawasaki, Japan), because it gives independent values of the three main VSCs, and can also contribute to discriminating between intraoral and extraoral origins [16]. Additionally, the identification and quantification of bacteria responsible for the production of VSCs may be useful in the diagnosis and management of halitosis.

Different mouthrinses try to reduce oral malodors, and a large number are on the market [17]. CHX is the most-used oral antiseptic [18] and as oral bacteria are the major source of intraoral halitosis, CHX is found in most mouthrinses for controlling halitosis [19]. However, as CHX is associated with reversible side effects such as tooth staining or taste alteration, products with lower percentages of CHX have been developed to avoid these disadvantages with similar efficacy, such as Halita™ (Dentaid, Cerdanyiola, Barcelona, Spain) mouthwash, which contains 0.05% CHX, 0.05% cetylpyridinium chloride (CPC), and zinc (0.14%) [20]. The combination of CHX and CPC has a synergistic effect on the reduction of VSC-producing bacteria [21,22,23,24] and plays a role in the decrease in halitosis in the long-term [25]. Moreover, zinc ions can bind to VSCs and transform them into nonodoriferous products [26], inducing short- and long-term neutralising effects [27,28,29].

There are no published studies on the efficacy against halitosis of Lacer Hali™, which consists of octenidine dihydrochloride (0.01%), zinc chloride (0.10%), sodium fluoride (0.55% for the toothpaste and 0.05% for the mouthwash), and xylitol (1%).

The aim of this study was to evaluate the efficacy of Lacer Hali™ (Lacer, Barcelona, Spain) mouthwash and toothpaste in subjects with intraoral halitosis after several applications under normal conditions of use.

## 2. Materials and Methods

### 2.1. Experimental Design

This clinical trial, with mouthrinse and toothpaste, was a single-center, randomized, double-blind, parallel trial with three arms, with, for each participant, an active duration of 18 days. This study followed the CONSORT (Consolidated Standards of Reporting Trials) [30] recommendation guidelines. An internal protocol designed by the researchers and Zurko (Zurko Research S.L., Madrid, Spain) was established to undertake the clinical trial. It was approved by the Ethics Committee of Rey Juan Carlos University (internal registration number: 2110202019020).

The study was designed by Zurko, and the tests were carried out under dental control in a dental clinic (Core Odontología y Estética Facial, S.L., Madrid, Spain).

### 2.2. Patient Population

The inclusion criteria to participate in the study were as follows: age 18–60 years, good health, and intraoral halitosis present at the start of the study. Participants must have had a score equal to or greater than 1 on the organoleptic scale. In addition, they must have had a score greater than 160 ppb for VSCs (obtained by the addition of hydrogen sulfide, methyl mercaptan, and dimethyl sulfide levels) and/or a score greater than 112 ppb for hydrogen sulfide and/or a score greater than 26 for methyl mercaptan. They must also have had one of these two elevated gases (described above), availability for the duration of the test, saliva rating greater than 0.2 mL/min, and have signed the informed consent form. The exclusion criteria were as follows: record of allergies or idiosyncrasies to tooth ingredients; had received treatment with antibiotics during the three weeks prior to the beginning of the study; had used commercialized anti-halitosis products up to 30 days before the test; had used another type of oral hygiene product for halitosis during the test; were pregnant or breastfeeding; were, after anamnesis, exploration, and panoramic radiography in the first clinical examination suspected to have sinus pathology, that could explain extraoral halitosis; and had extraoral halitosis, pseudohalitosis, and/or halitophobia.

Subjects were randomized to the tree arms using a list of random numbers by central telephone allocation through the Zurko Center: group 1 (experimental: Lacer Hali^TM^ mouthwash and toothpaste); group 2 (positive control: Halita^TM^ mouthwash and toothpaste); and group 3 (placebo: placebo mouthwash and toothpaste). The packaging and interior of the toothpastes and mouthwashes had the same visual appearance. The researcher who provided the materials to the participants was blinded.

The sample size was based on the study by Roldán S. et al. [21], where the effectiveness of Halita™ in two arms of 20 patients each was evaluated. A possible loss of participants of 15% was taken into account.

### 2.3. Patient Follow-Up

Clinical follow-up evaluations (T0, T1, T2, T3, and T4) were designed to evaluate the treatment efficacy over time and compared to the reference treatment and placebo. For these evaluations, the following analyses were conducted:

Organoleptic evaluation: A subjective assessment was performed according to the following scale [31]: 0 (absence of odor); 1 (barely perceived odor); 2 (slight odor but clearly perceived); 3 (moderate odor); 4 (strong odor); and 5 (extremely strong odor). Two dentists were olfactory-calibrated previously with the Smell Identification Test^TM^ but to reduce the risk of bias, all the organoleptic measurements were carried out by only one investigator (Laiqi Xiang).

The concentrations of VSCs (hydrogen sulfide, methyl mercaptan, and dimethyl sulfide) were determined using Oral Chroma™ (Abimedical, Kawasaki, Japan). We decided to focus on the hydrogen sulfide and methyl mercaptan in our analysis because they are the main contributors to intraoral halitosis [3,16,32].

The portable gas chromatograph Oral Chroma™ model CHM1 was checked and professionally calibrated by the manufacturers prior to the study.

#### 2.3.1. Baseline Visit

In the initial part of the first visit (T0), the patient was subjected to anamnesis, panoramic radiograph, saliva production rating, organoleptic evaluation, and VSCs determination. After 1 h (T1), the organoleptic test and VSCs determination were repeated. The products, instructions, and, follow-up table were given to the participants.

#### 2.3.2. Follow-Up Visits

After 7 (T2) and 15 days (T3) of continuous use of the treatment, the organoleptic evaluation, and VSC determination were repeated. At the last visit, the participants returned their products and they were given a placebo product. The spare products were weighed and the volume was measured to check the participant’s compliance.

#### 2.3.3. Final Visit (T4)

Three days after stopping the experimental treatment, the organoleptic evaluation, and VSCs determination were repeated. 

### 2.4. Procedure and Adverse Events 

Each participant was provided with a toothbrush, a tube of toothpaste, and mouthwash. The assigned products were applied twice a day for 15 days after the main meals (lunch and dinner). Participants applied a sufficient amount of toothpaste to cover the brush. They subsequently brushed their teeth and rinsed their mouth with 15 mL of mouthrinse for 1 min without rinsing the mouth later with water or another mouthrinse. All activities were reported in a patient record form.

On the T3 time point appointment, the participants were asked to return their products and new products containing the placebo were given to them. So, from T3 to T4 the participants of all groups were using the placebo. This was performed to test the duration of the therapeutic effect of the interventions once they were suppressed.

Participants were advised not to apply any other products orally during the study or receive any other dental treatments. At least one week before the clinic appointment, participants were advised to take the following precautions: abstain from alcohol in the last 12 h and avoid spicy foods, garlic, or onions two days before.

On the day of the appointment, the participants were asked to come with an empty stomach and to not use perfumed cosmetic products, but they were allowed to drink water up to 3 h before the appointment. As all the measurements were taken in the morning, they were also asked not to perform oral hygiene procedures from the previous day after dinner time No additional instructions or modifications to their routine hygiene techniques or eating habits were given. In cases of adverse reactions or doubt, the participants were advised to immediately suspend the application of the products, and to contact the research center.

### 2.5. Data Analysis 

Descriptive statistics of the organoleptic values and concentrations of VSCs, hydrogen sulfide, and methyl mercaptan were obtained at each of the study times. For each of the diagnostic tests, the percentage of variation relative to the baseline, placebo, and Halita^TM^ was calculated. Categorical models were adjusted to assess the differences between Lacer Hali^TM^ and placebo or Halita^TM^ over time. Generalized linear mixed-effect models (GLMM, negative-binomial model) were fitted to the gas concentration data to evaluate differences between Lacer Hali^TM^ and placebo or Halita^TM^ over time. When analyzing the organoleptic scale, generalized linear mixed models (GLMM, cumulative logit model) were fitted to the ordered categorical data to evaluate the differences between Lacerhali and the placebo or Halita^TM^ over time. To evaluate the efficacy of the product, the null hypothesis that there was no difference between Lacer Hali^TM^ and placebo or Halita^TM^ was evaluated by using Wald tests on the model parameters. R version 3.1.3 (R Core Development Team, R Foundation, Vienna, Austria) was used for the statistical analyses. The established significance value was 0.05 (95% confidence interval).

## 3. Results

Between June 2015 and November 2017, 69 participants were recruited and followed-up (Figure 1). The mean age of the participants was 38.39 (12.33) years, and 61.22% were women and 38.78% were men. The results by treatment groups were: placebo, with mean age of the participants was 38.39 (12.33) years, and 61.11% were women; Lacer Hali^TM^, with mean age of the participants was 38.91 (11.83) years, and 50.00% were women; Halita^TM^, with mean age of the participants was 38.77 (12.01) years, and 70.59% were women. During the study, there were five dropouts due to reasons unrelated to the participants, and in the data analysis, abnormal data were observed in four participants who were excluded. Each arm of the study consisted of 20 participants, so further recruitment was not considered.

### 3.1. Analysis of the Organoleptic Test

During the study, the most intense oral malodor was found in the placebo group. The average value for the Lacer Hali^TM^ group was always higher than that of the positive control group, (Halita^TM^, Figure 2). The mean values and percentage variation of organoleptic scores are represented in Table 1 and Table 2, respectively.

#### 3.1.1. Lacer Hali^TM^ vs. Placebo

The Lacer Hali^TM^ group showed a reduction in the average observed oral malodor at all study times compared to the placebo. After one hour of treatment (T1), there was a 17% reduction in the average oral malodor. This number was 33% at eight days (T2), 28%at 15 days (T3), and 19% after 18 days (T4) (Table 2).

Odor intensity with Lacer Hali^TM^ treatment was significantly lower (*p* = 0.009) after eight days (T2) of treatment, and close to statistically significant at 15 days (T3) (*p* = 0.059), compared to placebo treatment. There was a 92% and 83% probability of decrease of oral malodor with Lacer Hali^TM^ treatment compared to placebo treatment, respectively (Table 3).

#### 3.1.2. Lacer Hali^TM^ vs. Halita^TM^

There was an increase in the observed average oral malodor at all study times compared to the positive control treatment (Halita^TM^). After an hour of treatment (T1), there was a 4% increase in the average oral malodor, which was 3% at eight days (T2), 16% at 15 days (T3), and 10% after 18 days (T4), (Table 2).

With Lacer Hali^TM^ treatment, the odor intensity was not significantly greater or lower than the with Halita^TM^ treatment at any experimental time point (Table 3).

### 3.2. Analysis of Gases (VCS)

We analysed the values of hydrogen sulfide, methyl mercaptan, and the addition of the three gases (VCSs). The mean values and the percentages of variation of hydrogen sulfide, methyl mercaptan, and total VCSs are shown in Table 1 and Table 2, respectively. Figure 3 shows that Lacer Hali^TM^ had a tendency to decrease the hydrogen sulfide concentration at one hour (T1) compared with the placebo, with a statistically significant reduction at eight days (T2). However, a lower concentration of hydrogen sulfide was observed with Halita^TM^ compared with Lacer Hali^TM^ and the placebo at all study times.

As shown in Figure 4, Lacer Hali^TM^ showed a tendency to decrease the concentration of methyl mercaptan at one hour (T1) and at 15 days (T3) compared with placebo treatment. Regarding Halita^TM^ treatment, the tendency was to decrease the concentration of methyl mercaptan at all study times.

#### 3.2.1. Lacer Hali^TM^ vs. Placebo

There was a reduction in the concentration of hydrogen sulfide observed with Lacer Hali^TM^ after one hour (T1) and eight days (T2) of treatment, 51% and 49%, respectively (Table 1). There was a 57% probability on the eighth day (T2) of observing a significant reduction (*p* = 0.026) in hydrogen sulfide. At 15 (T3) and 18 days (T4) of treatment, the probabilities of finding a significantly higher concentration of hydrogen sulfide were 11% and 4%, respectively, but these results were not statistically significant (Table 4).

There was a reduction in the concentration of methyl mercaptan with Lacer Hali^TM^ after one hour (T1) and 15 days of treatment (T3), 38% and 17%, respectively (Table 1), but the results were not statistically significant. The concentration of methyl mercaptan with Lacer Hali^TM^ treatment was not significantly lower or higher than the concentration of methyl mercaptan with placebo treatment at any of the experimental time points (Table 4).

There was a reduction in the concentration of VSCs observed with Lacer Hali^TM^ after one hour (T1) and 8 days of treatment (T3), 47% and 23%, respectively (Table 2). The VSC concentration with Lacer Hali^TM^ treatment was not significantly lower or higher than the VSC concentration with placebo treatment at any of the experimental time points (Table 4).

#### 3.2.2. Lacer Hali^TM^ vs. Halita^TM^

Halita^TM^ was able to better reduce the concentration of hydrogen sulfide compared to Lacer Hali^TM^ at all experimentation time points (Table 2). At 15 days (T3), the concentration of hydrogen sulfide with Halita^TM^ was significantly lower than with Lacer Hali^TM^ treatment. Halita^TM^ showed a 64% probability of decreasing the hydrogen sulfide concentration (*p* = 0.013). After one hour (T1) of treatment, there was also a 36% decrease in the methyl mercaptan concentration (Table 2), but this result was not statistically significant. 

The concentration of methyl mercaptan with Lacer Hali^TM^ treatment was not significantly lower or higher than the concentration of methyl mercaptan with Halita^TM^ treatment at any of the experimental time points (Table 4).

After only one hour (T1) of Halita^TM^ treatment, there was a 22% decrease in the concentration of VSCs (Table 1), but this result was not statistically significant. At 15 days (T3), Halita^TM^ treatment decreased the concentration of VSCs more than Lacer Hali^TM^ treatment. Halita^TM^ showed a 64% and a 53% probability of decreasing the concentration of hydrogen sulfide (*p* = 0.013) and VCSs (*p* = 0.019), respectively (Table 4).

## 4. Discussion

This clinical trial with mouthrinse and toothpaste was designed to assess the effects of Lacer Hali^TM^ toothpaste and mouthrinse for which there are currently no scientific publications. The findings showed that its efficacy in reducing VSCs seemed to be higher than that of placebo treatment but no better than that of Halita^TM^ treatment.

Bad breath is a prevalent problem in many people worldwide, although some of those who have it may not be aware of it due to adaptation of their sense of smell [3]. Halitosis negatively affects the social interactions of a person’s daily life, causing discomfort and emotional stress. There are chemical methods, such as mouthrinse, and mechanical methods, such as brushing and flossing, to control oral malodors [24,33,34]. These therapeutic practices are complementary and used together, and represent the most effective approaches against halitosis. Multiple studies have evaluated the efficacy of an antimicrobial products using a single mouthrinse [35,36,37]. Other trial designs, such as ours, used both mechanical and chemical oral hygiene measures [35] in all participants.

CHX is the most common oral antiseptic, and its effectiveness for reducing of VSCs- producing bacteria has been demonstrated [19,22,38], but it is not exempt from side effects such as staining of the tongue and teeth, bad taste, and a reduction in taste perception [23,39]. Because of this, and to improve the antimicrobial effects on VSC-producing bacteria and consequently reduce oral malodors, new formulations based on a lower concentration of CHX associated to other antiseptics, such as cetylpyridinium chloride (CPC), have been developed. The combination of CHX/CPC/Zn (Halita™) appears to be the most effective to control the halitosis [19,21,25,40]. Zn also plays an important role as an odor-neutralizing agent because it is able to act on VSCs and transform them into nonodoriferous complexes [41,42,43,44,45,46,47].

Other therapeutic options should not be underestimated in the inhibition of oral malodors, including essential oils [48,49], triclosan [38], and oxidizing agents such as chlorine dioxide [35,50] and sodium chlorate [51]. Regarding this, octenidine may also be an additional therapeutic alternative, but there are no existing scientific studies about it. In this study, we compared Lacer Hali™, whose main composition is octenidine dihydrochloride, with a placebo group, and Halita™. The results obtained with Halita™ confirmed previous studies showing its greater efficacy against the placebo [52].

The organoleptic score scale was used to measure the intensity of halitosis due to its ease of use and frequent application in clinical trials to assess odor intensity in the oral cavity [20,35]. However, the results are based on an evaluator’s perception and could introduce bias. Therefore, as in other clinical trials [25,36,41,53], we used a portable gas chromatograph. Both Halimeter^TM^ (Interscan Corporation, Chatsworth, CA, USA) and OralChroma™ are available on the market to determine VSC concentrations. Halimeter^®^ measures the total amount of VSCs and only detects intraoral halitosis. OralChroma™ can differentiate between intraoral and extraoral halitosis, and provides separate values for each sulfide compound (hydrogen sulfide, methyl mercaptan, and dimethyl sulfide) [16,36]. In our study, although we used OralChroma™ and measured all three mentioned gases, we decided to focus our analysis on hydrogen sulfide and methyl mercaptan [54,55]. These two gases are the main contributors to intraoral malodor [3,16,32]. Additionally, methyl mercaptan and hydrogen sulfide are associated with bad odor of the tongue and periodontal pockets, respectively [56], whereas dymetil sulfide is more linked to extraoral causes of halitosis, such as respiratory, metabolic, or drug-associated [57], the menstrual cycle [58] or lifestyle factors such as ingestion of odiferous food [59].

The results obtained with OralChroma™ agree with those analyzed using the organoleptic scale. Several studies have demonstrated a high correlation between organoleptic scores and VSCs measurements obtained by a sulfide monitor or a gas chromatograph, showing that both methods are suitable to diagnose halitosis [45,60]. According to this, in our study, Lacer Hali^TM^ was demonstrated to be significantly better than the placebo at T2. Moreover, a decrease in both organoleptic scores and hydrogen sulfide levels was detected. For the rest of the time, there was no statistically significant difference between Lacer Hali^TM^ and the placebo. However, we can appreciate a minor trend in the decrease of the total VSCs measurement in comparison with the organoleptic results. This finding suggests that other odorous nonsulfur compounds may contribute to oral cavity malodor. Thus, indole, skatole, cadaverine, putrescine, and short-chain fatty acids may play a role in the genesis of halitosis [61,62,63]. On the other hand, although we observed that the placebo seems to reduce methyl mercaptan levels, the difference was not statistically significant and we assume it could be due to chance.

In our study, there were hardly any dropouts (9 of 69), and none of the cases of dropout were due to the development of unpleasant sensations or adverse effects. In our clinical trial, a complete follow-up of three arms with 20 patients each was performed based on a clinical trial conducted with Halita^TM^ [21], where each of the two arms of the study had 20 participants. Other trial designs, have been performed with Halita^TM^, where its efficacy was compared against four groups, and each of the arms consisted of 18 patients [25]; therefore, the sample size of our study could be considered adequate.

The tongue is where most of the bacteria in the mouth are located and is mostly responsible for producing the bad smells. Some studies have incorporated tongue cleaning procedures in their designs, with no beneficial effects on the microbiota [46], although they do reduce the organoleptic measurement index and the tongue coating index [64]. On the other hand, several investigations have described that the major source of VCSs, especially hydrogen sulfide, is the lingual dorsum [65,66,67]. Therefore, LacerHali™ seems to be more efficient for hydrogen sulfide control than for methyl mercaptan control. This suggests that Lacer Hali™ could be a good therapeutic option as a complement of mechanical cleaning for the management of intraoral halitosis, whose prevalent origin is the lingual dorsum.

From T3 to T4 time points the participants of all groups used placebo products in order to check the persistence of the therapeutic effects once the use of the products was suspended. As was expected, the levels of organoleptic scores and hydrogen sulfide increased in the three groups in this period. However, whereas the levels of methyl mercaptan rose for the Lacer Hali^TM^ and placebo group, these decreased for the Halita^TM^ group. These results can suggest that Halita^TM^, due to its chlorhexidine content, had a residual antimicrobial effect (substantivity) that could persist over the time [68,69].

Regarding oral hygiene measures, in some designs [21], the participants were asked to modify their daily cleaning behaviors. In this clinical trial, they were recommended to continue with their usual brushing technique and usual behavior to minimize the risk of bias.

It would have been interesting to include an analysis to quantify odoriferous microorganisms using a polymerase chain reaction (PCR) technique [70,71,72]. However, we consider that combined analysis using organoleptic measurements and VSC analysis is sufficient to test the overall efficacy of Lacer Hali™. 

Concerning the study population selection, in our trial we decided to establish the cut-off point at ≥1 and ≥160 ppb for the organoleptic score and VSC concentrations, respectively. Similar values were used in previous studies [21,25,40]. In relation to this, there is a strong heterogeneity in the literature about the threshold of inclusion criteria for the organoleptic score and VSC concentration, being from 1 to 4 and 80 to 300 ppb, respectively [19,24].

It would have been rewarding for our study to have performed additional measurements of nasal breath air, blood analysis, and metabolic and digestive system tests in order to completely discard extraoral halitosis pathologies, but performing these tests exceeded the resources of this study.

Our main goal was to investigate the effects of our intervention in terms of reduction or increase of intraoral halitosis, although a complete elimination of oral malodor was not possible independently of its specific oral origin. Regarding this, from a clinical point of view, it is important for the success of halitosis treatments to identify the source of oral malodor to establish the suitable intervention. Intraoral halitosis caused by periodontitis, gingivitis, lingual dorsum, or other intraoral conditions will require specific additional treatments. It would be interesting for future studies to investigate the effect of this mouthrinse and toothpaste in volunteers with intraoral halitosis classified according to its prevalent origin (e.g., lingual dorsum, gingivitis, or periodontitis). Likewise, it would be clinically useful to determine the effectiveness of this novel mouthrinse depending on the severity of halitosis. Additionally, long-term studies that incorporate identification and quantification techniques for microorganisms are needed to assess the effect of octenidine on microbiota levels in tongue and periodontal localizations, as well as its correlation with VSC levels and organoleptic scores.

In conclusion, although more studies are needed, the results of this clinical trial show that the combination of Lacer Hali™ toothpaste and mouthrinse could be considered an alternative to CHX-based products in the management of halitosis.

## Figures and Tables

**Figure 1 jcm-10-02256-f001:**
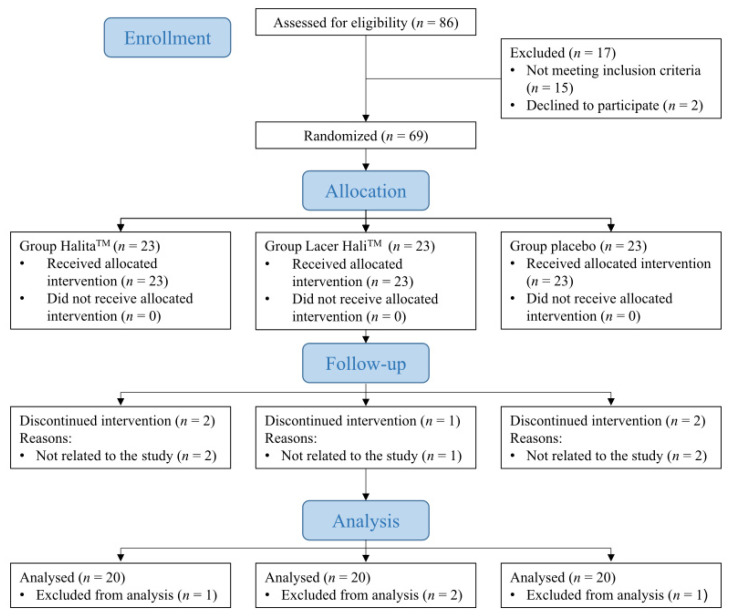
CONSORT flow chart of the study patients.

**Figure 2 jcm-10-02256-f002:**
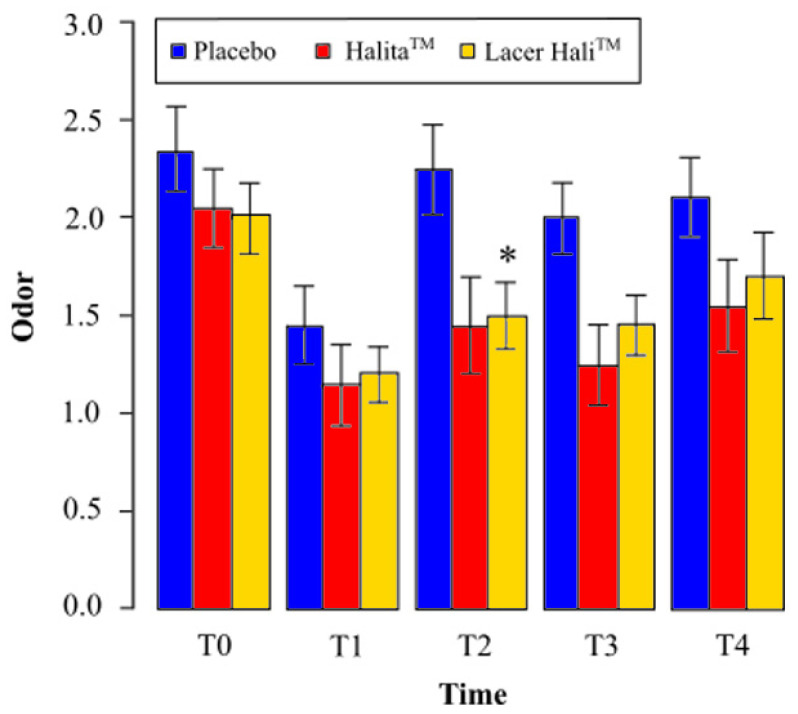
Mean perceived odor and standard deviation at different times for the organoleptic test. *: Statistically significant results (*p* < 0.05) between the placebo and Lacer Hali^TM^ groups were found. No significant differences were found between the Halita^TM^ and Lacer Hali^TM^ groups.

**Figure 3 jcm-10-02256-f003:**
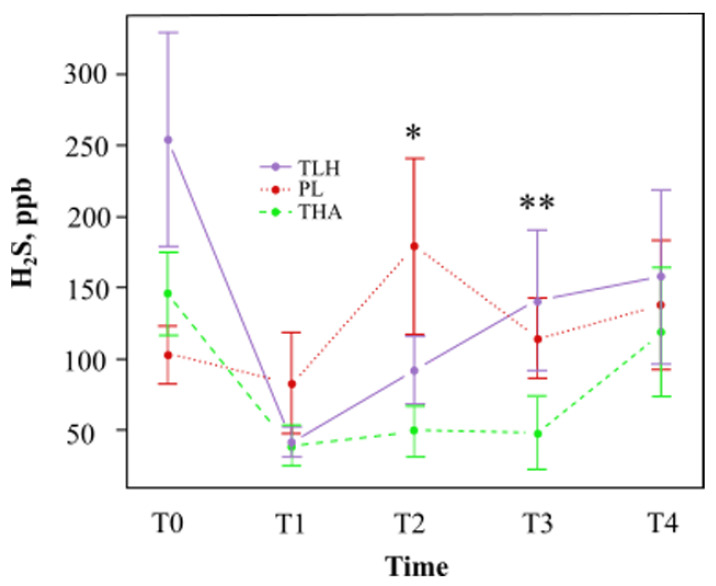
Mean average hydrogen concentration (parts per billion) and standard deviation at different time points (TLH, Lacer Hali^TM^ treatment; PL, placebo; THA, Halita^TM^ treatment). *: Statistically significant results (*p* < 0.05) between the placebo and Lacer Hali^TM^ groups were found. **: Statistically significant results (*p* < 0.05) between the Lacer Hali^TM^ and Halita^TM^ groups were found.

**Figure 4 jcm-10-02256-f004:**
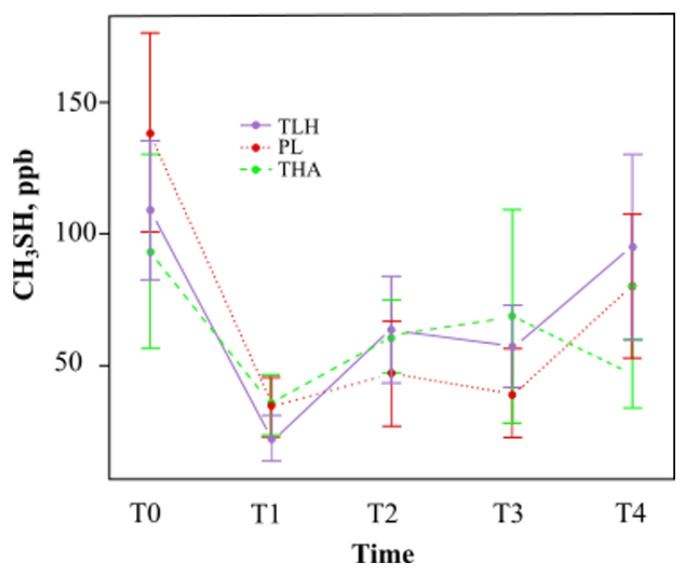
Mean methyl mercaptan concentration (parts per billion) and standard deviation at different times (TLH, Lacer Hali^TM^ treatment; PL, placebo; THA, Halita^TM^ treatment). There were no statistically significant results.

**Table 1 jcm-10-02256-t001:** Descriptive statistics of the evaluation of odors and the concentrations of hydrogen sulfide, methyl mercaptan, and VSCs.

		Organoleptic		Hydrogen Sulfide		Methyl Mercaptan		VSC	
Stage	Group	Mean	SD	Mean	SD	Mean	SD	Mean	SD
T0	Lacer Hali^TM^	2.00	0.79	253.75	335.55	108.60	117.30	383.30	433.92
	Placebo	2.35	0.93	102.15	91.07	93.20	165.15	223.50	179.95
	Halita^TM^	2.05	0.89	145.20	131.27	138.30	168.98	322.65	265.32
T1	Lacer Hali^TM^	1.20	0.62	40.50	48.16	22.10	38.11	74.30	95.17
	Placebo	1.45	0.89	82.00	157.82	35.70	48.13	139.25	176.29
	Halita^TM^	1.15	0.93	37.75	61.34	34.30	49.53	94.70	128.32
T2	Lacer Hali^TM^	1.50	0.76	91.20	108.11	63.45	91.17	201.75	209.89
	Placebo	2.25	1.02	178.65	275.70	60.90	61.61	263.45	330.27
	Halita^TM^	1.45	1.10	49.10	81.56	46.90	90.27	104.80	169.93
T3	Lacer Hali^TM^	1.45	0.69	140.70	221.21	57.10	69.61	218.20	271.59
	Placebo	2.00	0.79	113.80	124.66	68.65	181.20	198.60	206.33
	Halita^TM^	1.25	0.91	47.25	113.92	39.25	74.98	102.40	184.99
T4	Lacer Hali^TM^	1.70	0.98	157.15	274.29	95.00	156.67	264.35	395.54
	Placebo	2.10	0.91	137.10	203.13	46.50	57.17	192.20	252.78
	Halita^TM^	1.55	1.05	118.15	201.71	80.10	121.86	213.60	305.49

SD, standard deviation; VSCs, volatile sulfur compounds.

**Table 2 jcm-10-02256-t002:** Variations in odor intensity and the concentrations of hydrogen sulfide, methyl mercaptan, and total VSCs.

		Organoleptic			Hydrogen Sulfide			Methyl Mercaptan			VSC		
Stage	Group	% of Variation Relative to Baseline	% of Variation Relative to Placebo	% of Variation Relative to Halita^TM^	% of Variation Relative to Baseline	% of Variation Relative to Placebo	% of Variation Relative to Halita^TM^	% of Variation Relative to Baseline	% of Variation Relative to Placebo	% of Variation Relative to Halita^TM^	% of Variation Relative to Baseline	% of Variation Relative to Placebo	% of Variation Relative to Halita^TM^
T0	Lacer Hali^TM^	--	−15%	−2%	--	148%	75%	--	17%	−21%	--	71%	19%
	Placebo	--	--	--	--	--	--	--	--	--	--	--	--
	Halita^TM^	--	--	--	--	--	--	--	--	--	--	--	--
T1	Lacer Hali^TM^	−40%	−17%	4%	−84%	−51%	7%	−80%	−38%	−36%	−81%	−47%	−22%
	Placebo	−38%	--	--	−20%	--	--	−62%	--	--	−38%	--	--
	Halita^TM^	−44%	--	--	−84%	--	--	−75%	--	--	−71%	--	--
T2	Lacer Hali^TM^	−25%	−33%	3%	−64%	−49%	86%	−42%	4%	35%	−47%	−23%	93%
	Placebo	−4%	--	--	75%	--	--	−35%	--	--	18%	--	--
	Halita^TM^	−29%	--	--	−64%	--	--	−66%	--	--	−68%	--	--
T3	Lacer Hali^TM^	−28%	−28%	16%	−45%	24%	198%	−47%	−17%	45%	−43%	10%	113%
	Placebo	−15%	--	--	11%	--	--	−26%	--	--	−11%	--	--
	Halita^TM^	−39%	--	--	−45%	--	--	−72%	--	--	−68%	--	--
T4	Lacer Hali^TM^	−15%	−19%	10%	−38%	15%	33%	−13%	104%	19%	−31%	38%	24%
	Placebo	−11%	--	--	34%	--	--	−50%	--	--	−14%	--	--
	Halita^TM^	−24%	--	--	−38%	--	--	−42%	--	--	−34%	--	--

VSC_s_,: volatile sulfur compounds; --,not applicable.

**Table 3 jcm-10-02256-t003:** A cumulative logit mixed model was fitted to evaluate the difference between Lacer Hali^TM^ vs. Halita^TM^ and Lacer Hali^TM^ vs. placebo treatments at different time points during the organoleptic evaluation. Statistically significant *p*-value < 0.05.

Stage	Group	Predicted Probability	z	*p*-Value
T0	Lacer Hali^TM^ vs. Placebo	−61%	−0.98	0.326
	Lacer Hali^TM^ vs. Halita^TM^	−25%	−0.43	0.667
T1	Lacer Hali^TM^ vs. Placebo	−58%	−0.92	0.360
	Lacer Hali^TM^ vs. Halita^TM^	29%	0.39	0.694
T2	Lacer Hali^TM^ vs. Placebo	−92%	−2.60	0.009
	Lacer Hali^TM^ vs. Halita^TM^	19%	0.23	0.816
T3	Lacer Hali^TM^ vs. Placebo	−83%	−1.89	0.059
	Lacer Hali^TM^ vs. Halita^TM^	56%	0.94	0.345
T4	Lacer Hali^TM^ vs. Placebo	−72%	−1.35	0.177
	Lacer Hali^TM^ vs. Halita^TM^	42%	0.58	0.563

**Table 4 jcm-10-02256-t004:** Cumulative logit mixed model fitted to evaluate the difference between Lacer Hali^TM^ vs. Halita^TM^ and Lacer Hali^TM^ vs. placebo at different time points in the evaluation of the concentration of hydrogen sulfide, methyl mercaptan, and VSCs.

Stage	Group	Predicted Probability	z	*p*-Value
T0	Hydrogen sulfide			
	Lacer Hali^TM^ vs. Placebo	46%	1.72	0.0860
	Lacer Hali^TM^ vs. Halita^TM^	30%	1.03	0.3010
	Methyl mercaptan			
	Lacer Hali^TM^ vs. Placebo	10%	0.32	0.7470
	Lacer Hali^TM^ vs. Halita^TM^	−15%	−0.48	0.6320
	VSC			
	Lacer Hali^TM^ vs. Placebo	25%	1.05	0.2945
	Lacer Hali^TM^ vs. Halita^TM^	3%	0.12	0.9050
T1	Hydrogen sulfide			
	Lacer Hali^TM^ vs. Placebo	−31%	−0.90	0.3673
	Lacer Hali^TM^ vs. Halita^TM^	16%	0.40	0.6880
	Methyl mercaptan			
	Lacer Hali^TM^ vs. Placebo	−49%	−1.55	0.1203
	Lacer Hali^TM^ vs. Halita^TM^	−42%	−1.26	0.2070
	VSC			
	Lacer Hali^TM^ vs. Placebo	−40%	−1.56	0.1190
	Lacer Hali^TM^ vs. Halita^TM^	−5%	−0.15	0.8810
T2	Hydrogen sulfide			
	Lacer Hali^TM^ vs. Placebo	−57%	−2.23	0.0256
	Lacer Hali^TM^ vs. Halita^TM^	37%	1.07	0.2852
	Methyl mercaptan			
	Lacer Hali^TM^ vs. Placebo	−11%	−0.31	0.7560
	Lacer Hali^TM^ vs. Halita^TM^	26%	0.76	0.4480
	VSC			
	Lacer Hali^TM^ vs. Placebo	−28%	−1.09	0.2778
	Lacer Hali^TM^ vs. Halita^TM^	40%	1.58	0.1142
T3	Hydrogen sulfide			
	Lacer Hali^TM^ vs. Placebo	11%	0.30	0.7630
	Lacer Hali^TM^ vs. Halita^TM^	64%	2.49	0.0130
	Methyl mercaptan			
	Lacer Hali^TM^ vs. Placebo	22%	0.62	0.5350
	Lacer Hali^TM^ vs. Halita^TM^	32%	0.97	0.3340
	VSC			
	Lacer Hali^TM^ vs. Placebo	−1%	−0.04	0.9673
	Lacer Hali^TM^ vs. Halita^TM^	53%	2.35	0.0190
T4	Hydrogen sulfide			
	Lacer Hali^TM^ vs. Placebo	4%	−0.11	0.9150
	Lacer Hali^TM^ vs. Halita^TM^	16%	0.46	0.6440
	Methyl mercaptan			
	Lacer Hali^TM^ vs. Placebo	8%	0.20	0.8380
	Lacer Hali^TM^ vs. Halita^TM^	−21%	−0.62	0.5330
	VSC			
	Lacer Hali^TM^ vs. Placebo	−1%	−0.03	0.9790
	Lacer Hali^TM^ vs. Halita^TM^	1%	−0.03	0.9756

Statistically significant *p*-value < 0.05. VSCs, volatile sulfur compounds.

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
