# Peer review of "Evaluation of the Efficacy of Lacer HaliTM Treatment on the Management of Halitosis: A Randomized Double-Blind Clinical Trial"

_jcm, 2021, doi:10.3390/jcm10112256_

Round 1
Reviewer 1 Report
The manuscript is well written and it adds new informations for international literature.
In this study the authors evaluate the efficacy of LacerhaliTM mouthwash and toothpaste in subjects with intraoral halitosis after several applications under normal conditions of use. The manuscript adds new informations to international literature but can be improved. The references could be up to date. Table I should be rewritten because it is unclear.
Author Response
The manuscript is well written and it adds new informations for international literature. In this study the authors evaluate the efficacy of LacerhaliTM mouthwash and toothpaste in subjects with intraoral halitosis after several applications under normal conditions of use. The manuscript adds new informations to international literature but can be improved. The references could be up to date. Table I should be rewritten because it is unclear.
Reply: We are very thankful for your effort and time that you invested to improve the manuscript. Following your suggestions we have updated the references including the following ones:
- Mendes, L.; Coimbra, J.; Pereira, A.L.; Resende, M.; Pinto, M.G. Comparative effect of a new mouthrinse containing chlorhexidine, triclosan and zinc on volatile sulphur compounds: a randomized, crossover, double-blind study. Int. J. Dent. Hyg. 2016, 14, 202–208, doi:10.1111/idh.12132.
- Jervøe-Storm, P.-M.; Schulze, H.; Jepsen, S. A randomized cross-over short-term study on the short-term effects of a zinc-lactate containing mouthwash against oral malodour. J. Breath Res. 2019, 13, 26005, doi:10.1088/1752-7163/aaf401.
- Satthanakul, P.; Taweechaisupapong, S.; Paphangkorakit, J.; Pesee, M.; Timabut, P.; Khunkitti, W. Antimicrobial effect of lemongrass oil against oral malodour micro-organisms and the pilot study of safety and efficacy of lemongrass mouthrinse on oral malodour. J. Appl. Microbiol. 2015, 118, 11–17, doi:10.1111/jam.12667.
- Alsaffar, D.; Alzoman, H. Efficacy of antioxidant mouthwash in the reduction of halitosis: A randomized, double blind, controlled crossover clinical trial. J. Dent. Sci. 2021, 16, 621–627, doi:10.1016/j.jds.2020.10.005.
- Ye, W.; Zhang, Y.; He, M.; Zhu, C.; Feng, X.-P. Relationship of tongue coating microbiome on volatile sulfur compounds in healthy and halitosis adults. J. Breath Res. 2019, 14, 16005, doi:10.1088/1752-7163/ab47b4.
- Seerangaiyan, K.; Jüch, F.; Winkel, E.G. Tongue coating: its characteristics and role in intra-oral halitosis and general health-a review. J. Breath Res. 2018, 12, 34001, doi:10.1088/1752-7163/aaa3a1.
Furthermore, we have restructured the Table I. For a better understanding we have made two independent tables (tables 1 and 2).

Reviewer 2 Report
The authors have carried out an interesting and well-reported study on the efficacy with Lacerhali Treatment against halitosis. This is a randomized double-blind clinical trial based on a timely topic.
The authors have well investigated on novel mouthwash and toothpaste in subjects with halitosis under normal conditions of use.
The main issue is related to the introduction: it is not properly described and with a superficial approach on several related topics.
The authors should briefly discuss the impact of infections and inflammation on oral tissues and their clinical applications (e.g. Please, see and discuss: “Marrelli M, et al. Oral infection by Staphylococcus aureus in patients affected by White Sponge Nevus: a description of two cases occurred in the same family. Int J Med Sci. 2012;9(1):47-50.” – AND – “Inchingolo F, et al. Oral piercing and oral diseases: a short time retrospective study. Int J Med Sci. 2011;8(8):649-52. doi: 10.7150/ijms.8.649.”).
Moreover, other important causes of halitosis could be related to severe pathologies, such as cancer (Please, discuss “Inchingolo F, Tatullo M, Abenavoli FM, Marrelli M, Inchingolo AD, Inchingolo AM, Dipalma G. Non-Hodgkin lymphoma affecting the tongue: unusual intra-oral location. Head Neck Oncol. 2011 Jan 4;3:1.”) and they must be recognized before any therapeutic plan.
Minor suggestion:
Finally, Conclusions should be improved with clear take-home messages.
Author Response
Reviewer 2
The authors have carried out an interesting and well-reported study on the efficacy with Lacerhali Treatment against halitosis. This is a randomized double-blind clinical trial based on a timely topic.
The authors have well investigated on novel mouthwash and toothpaste in subjects with halitosis under normal conditions of use.
The main issue is related to the introduction: it is not properly described and with a superficial approach on several related topics.
The authors should briefly discuss the impact of infections and inflammation on oral tissues and their clinical applications (e.g. Please, see and discuss: “Marrelli M, et al. Oral infection by Staphylococcus aureus in patients affected by White Sponge Nevus: a description of two cases occurred in the same family. Int J Med Sci. 2012;9(1):47-50.” – AND – “Inchingolo F, et al. Oral piercing and oral diseases: a short time retrospective study. Int J Med Sci. 2011;8(8):649-52. doi: 10.7150/ijms.8.649.”).
Moreover, other important causes of halitosis could be related to severe pathologies, such as cancer (Please, discuss “Inchingolo F, Tatullo M, Abenavoli FM, Marrelli M, Inchingolo AD, Inchingolo AM, Dipalma G. Non-Hodgkin lymphoma affecting the tongue: unusual intra-oral location. Head Neck Oncol. 2011 Jan 4;3:1.”) and they must be recognized before any therapeutic plan.
Reply: We are very grateful for your time and effort to improve our manuscript. Following your suggestions we have changed the introduction emphasising the role of halitosis as manifestation of certain oral soft tissue conditions and added the proposed references. You can find the changes made to the script with the track changes function.
Minor suggestion: Finally, Conclusions should be improved with clear take-home messages.
Reply: Likewise, we have modified the conclusion paragraph following your recommendation.

Reviewer 3 Report
Please find my comments in a separate file.

Author Response
We are extremely grateful for the time and effort that you have invested in order to improve our manuscript. According to your recommendations, an extensive and intensive English language and style correction were made in the all text. It was also reviewed by a medical English reviewer. We appreciate your suggestions and the answerers to your questions are below:
Please be clear throughout the manuscript that the intervention includes a mouth rinse and toothpaste.
Reply: We have clarified this point adding mouth rinse and toothpaste every time we refer the intervention in the manuscript.
It is doubtful if chlorhexidine can be considered as gold standard in treatment of intra-oral halitosis. Please add a reference.
Decreased VSC values were found but what did the persons have halitosis?
Reply: There is a general trend in the decrease of VSC of Lacer HaliTM compared to placebo, which is statistically significant for hydrogen sulphide and organoleptic scores at T2.
The tongue biofilm (Roldan et al 2003, Washio J et al 2005, Ye W 2019, Seerangaiyan K 2018) is mainly associated to the production of hydrogen sulphide and considering the lingual dorsum as the prevalent cause of intraoral halitosis (Tangerman A et al 2002, Richter J L et al 1996, Loesche WJ et al 2002, Wei et al 2019), is highly probable that our intervention could be effective to reduce the participant halitosis.
Page 2, first paragraph: halitosis is not diagnosed on basis of microbial parameters. However, microbial diagnostics may be an addition to diagnostics of halitosis. Please adjust.
Reply: It is absolutely correct that microbial parameters can be helpful in the management of oral malodour but they per se are not sufficient to the diagnostic of halitosis. We have clarified this point in the text.
Please define the placebo mouth rinse and toothpaste. Halita does not consist only chlorhexidine. This should be reported and discussed. Also other active compounds may play a role in treatment with halitosis such cetylpyridinium chloride and zinc. Please also add these important compounds to the introduction and discussion.
Reply: Placebo mouthrinse contains: sodium methyl hydroxybenzoate 0,1g; sodium propydroxybenzoate 0,06g and excipient c.s.p. 100mL. Placebo toothpaste contains: sodium monofluorophosphate 0,38g; preservative: sodium methyl hydroxybenzoate 0,2g and excipient c.s.p. 100g.
Thank you for your suggestion to enhance our introduction and discussion. We have discussed, in the introduction and discussion section, the important role of CPC and Zn in the management of halitosis and added references about it.
Has the study been registered in the clinical trial register? What is the number of the registration?
Reply: According to the Spanish law (Royal Decree 1090/201, December 4) that regulates clinical trials with drugs, the Ethic Committees for Research with drugs, and the Spanish Registry of Clinical Studies (Spain), the registration of clinical trials without products not considered as drugs (i. e. cosmetics) is voluntary. Lacer HaliTM is classified in the category of “cosmetic” products. So, our clinical trial was not registered in a public database. Anyway, the study was registered internally in Zurko Research as 03/HAL_077_15-001.
Why did you use the threshold of 160ppb for inclusion? And how did you calculate it? Please add definitions and references for diagnosis.
Reply: Baharvant et al (2008) defined the values between 160 and 250 ppb as weak halitosis, which is correlated with our other inclusion criteria (organoleptic score greater than 2). The systematic reviews of Blom et al 2012 and Slot et al 2015 have reported a substantial heterogeneity of the participant inclusion criteria in previous clinical trials about the efficacy of mouth rinses, being the range of VSC between 80 (Rassameemasmaung et al 2007) and 300 ppb (Boyd et al 2007).
The other studies carried out with a similar design to ours established the cut-off point between 150 (Dadamio et al 2013) and 170 (Roldán and Winkel 2003) ppb as the inclusion criterion of the volunteers. Thus, we decided to set our limit to 160, considering it as the middle figure between both.
How did you distinct between different forms of halitosis?
Reply: The aim of our investigation was to test the effectiveness of a novel treatment (Lacer Hali TM mouth rinse and toothpaste in the management of intraoral halitosis. Due to this, or study population should be volunteers with intraoral halitosis. In order to exclude the extraoral causes of halitosis, we established a exhaustive anamnesis (specially the presence of respiratory, metabolic, digestive disorders and the ingestion of some drugs), intraoral exploration and finally, a panoramic RX (to check the sinus status).
Is known that hydrogen sulphide and methyl mercaptan are the predominant cause of oral malodour whereas dimethyl sulphide is more correlated to extraoral or blood borne halitosis (Tangerman et al 2002 Int Dent J, Tangerman et al 2007 JCP, Tangerman and Winkel 2008 J Breath Research). The requirements for he participants were that when the total value of VSC overscored 160ppb, hydrogen sulphide and/or methyl mercaptan also must be above 112 and 26 ppb respectively. Thus, we can ensure the main intraoral origin of the halitosis of the volunteers.
How many dentist did perform the organoleptic testing? How were they calibrated?
Reply: Two trained evaluators were initially calibrated by using Smell Identification Test TM. Nevertheless, in order to decrease the risk of bias, all the organoleptic measurements were taken only by one dentist. We clarify this point in the manuscript.
Which model of OralChroma was used?
Reply: Oral Chroma model CHM1
Description of the study cohorts is missing. Did the participants have gingivitis? What about their plaque and bleeding scores?
Reply: The cohort population of our study are persons with healthy general status and intraoral halitosis.
Our sample is made up of participants with intraoral halitosis, regardless the origin of that oral malodour, so we cannot ensure the gingival status of the volunteers.
Even though it would have been desirable and helpful for the diagnosis to measure the plaque and bleeding index, we did not performed these scores because the aim of our trial was to determinate the efficacy of Lacer HaliTM on persons with intraoral halitosis, regardless the cause of the intraoral malodour. We consider that the plaque and bleeding scores could be useful to discriminate among different types and severity of intraoral halitosis but it is not necessary to distinguish between intraoral and extraoral halitosis. However, the registration of those scores could be interesting for future investigations addressed to differentiate the efficacy of these products depending on the origin of intraoral halitosis (e.g., gingivitis, periodontitis and lingual dorsum bacteria accumulation).
We appreciate your suggestion and we have decided to include this point in the discussion.
The intervention products were used after lunch and dinner. Most individuals use clean their teeth/mouth in the morning and evening. How did you control for possible additional cleaning moments and variation in these measures? How did control for compliance?
Reply: The participants signed a compliance form where they committed to comply the study instructions. Furthermore, volunteers were asked to return the spare products and those were weighed and measured.
What time of the day did you perform the breath odor measurements?
Reply: All the measurements were performed in the morning. We include this point in the manuscript.
Please provide more information about the statistical analysis.
Reply: More information regarding the statistical tests carried out has been included in the statistical analysis section.
Figure 1 shows number that are not clear; if 2 out of 21 individuals were excluded you end up with 19 persons for that group. Please adjust and clarify.
Reply: Thank you for identifying the error in the flow chart. The number of patients per group is correct. The error in the figure has been corrected.
Figure 2 is missing p-values.
Reply: The p-values of the tests with significant results have been included in Figure 2. You can see from the legend that the asterisk indicates that there is a significant result of the statistical test.
Why don’t you report concentrations of all 3 detected compounds of OralChroma?
Reply: Effectively, the readings of oral chroma show the concentration of hydrogen sulphide, methyl mercaptan and dimethyl sulphide. Considering that the main contributor to intraoral malodour are hydrogen sulphide and methyl mercaptan and, to a lower extend, dimethyl sulphide (Tangerman A 2002, Tangerman A and Winkel E G 2007, Tangerman A and Winkel E G 2008), we decided to focus on the first two gases for our analysis as in the study of Wilhelm D et al 2010. Furthermore, dimethyl sulphide may be associated to several extraoral causes as respiratory, metabolic, the ingestion of some drugs (Murata et al 2003,) or lifestyle (ingestion of odoriferous foods (Suarez et al 1999), the menstrual cycle (Kawamoto et al 2010), etc).
We have added this clarification in the section 2.3 of material and methods, and in the discussion we address the role of dimethyl sulphide in extraoral halitosis.
Figure 4. Placebo seems to give a lower MM values. Is this correct?
Reply: Yes, the values of MM are correct. Surprisingly, the placebo shows the lower MM values, but this decrease is not statically significant. So, it is highly probable that could be due to chance. We discuss about this in the discussion section.
Please discuss critically the limitations of the study.
Reply: According to your recommendations, we have rewritten the discussion section including more points about the limitations and improvements that could have been made to the study.

Round 2
Reviewer 2 Report
revisions addressed
Author Response
Thank you for your work to improve our manuscript. We found your suggestions very useful.
Kind regards.
Reviewer 3 Report
Please find my comments in an attachment.

Author Response
Thank you for your work to improve our manuscript. You can find the answers in the attached document.
Kind regards.
